# Multiparameter squeezing for optimal quantum enhancements in sensor networks

Manuel Gessner [1✉], Augusto Smerzi[2] & Luca Pezzè[2]

Squeezing currently represents the leading strategy for quantum enhanced precision measurements of a single parameter in a variety of continuous- and discrete-variable settings and technological applications. However, many important physical problems including imaging and field sensing require the simultaneous measurement of multiple unknown parameters. The development of multiparameter quantum metrology is yet hindered by the intrinsic difficulty in finding saturable sensitivity bounds and feasible estimation strategies. Here, we derive the general operational concept of multiparameter squeezing, identifying metrologically useful states and optimal estimation strategies. When applied to spin- or continuous-variable systems, our results generalize widely-used spin- or quadrature-squeezing parameters. Multiparameter squeezing provides a practical and versatile concept that paves the way to the development of quantum-enhanced estimation of multiple phases, gradients, and fields, and for the efficient characterization of multimode quantum states in atomic and optical sensor networks.

[1] Laboratoire Kastler Brossel, ENS-PSL Université, CNRS, Sorbonne Université, Collège de France, 24 Rue Lhomond, 75005 Paris, France. [2] QSTAR, CNR-INO and LENS, Largo Enrico Fermi 2, 50125 Firenze, Italy. ✉email: manuel.gessner@ens.fr

Squeezing of quantum observables is a central strategy to improve measurement sensitivities beyond classical limits and has thus become a key concept in quantum metrology, leading to major theoretical and experimental advancements in the field[1–6]. Furthermore, squeezing is a convenient approach to witness genuine quantum properties such as entanglement[7,8] or nonclassicality[9], only requiring knowledge of first and second moments of suitable linear observables that can be obtained experimentally with high efficiency. The concept of squeezing is most useful for the important class of Gaussian states that is routinely generated in atomic and photonic experiments[1,10–13].

While well understood in the framework of single-parameter estimation[1–5], the existing notion of squeezing is insufficient to characterize the sensitivity of multiparameter estimation. Indeed, the simultaneous estimation of several parameters can be more efficient than the optimal estimation of each parameter separately[14–17]. This interesting prediction is under intensive investigation[18–23] and can revolutionize many technological applications such as quantum imaging[24], microscopy and astronomy[25–28], sensor networks[15,16,23], and atomic clocks[29], by enhancing the estimation sensitivity of inhomogeneous intensity distributions, vector fields, and gradients[30–34]. However, the current framework of multiparameter quantum metrology has developed based on the notion of the quantum Fisher information matrix[35]: a figure of merit that is not straightforward to extract experimentally and is also generally hard to determine theoretically. Furthermore, the sensitivity limit defined by the inverse of the quantum Fisher information matrix, namely, the multiparameter quantum Cramér–Rao bound[35], is, in general, not saturable[36,37]. Alternative approaches based on the Holevo bound are in principle asymptotically saturable but require, in general, complex measurements on multiple copies of the state[38–42].

In this work, we introduce the general notion of metrological multiparameter squeezing for continuous and discrete variables. This concept follows directly from a specific operational approach to multiparameter estimation based on mean values and variances of the measured observables. Metrological multiparameter squeezing thus provides an accessible and saturable lower bound to the quantum Fisher matrix that is tight for the broad and experimentally relevant class of Gaussian states. We further use matrix order inequalities to analytically optimize the measurement observables as a function of accessible observables. Our framework is neither limited to specific systems nor to a particular class of observables and provides an efficient characterization of useful quantum resources for multiparameter estimation for any given set of commuting observables that are simultaneously measured. For linear spin observables, our method gives rise to the spin-squeezing matrix as a natural generalization of the spin-squeezing coefficient introduced by Wineland et al.[5] to multiparameter settings. The spin-squeezing matrix reveals the role of nonlocal squeezing, i.e., squeezing in a nonlocal superposition of modes for simultaneous estimations of multiple parameters that can enhance the sensitivity of specific linear combinations of parameters. We further identify optimal strategies for displacement sensing in continuous variables, where nonlocal squeezing over $M$ modes can reduce the estimation error up to a factor $\sqrt{M}$. To address the properties of non-Gaussian states, we demonstrate that our approach can yield a multiparameter sensitivity as large as the classical Fisher matrix (and even the quantum Fisher matrix, whenever the multiparameter quantum Cramér–Rao bound is saturable).

## Results

**Multiparameter method of moments.** In multiparameter quantum metrology[35] the goal is to estimate a family of unknown parameters $\boldsymbol{\theta} = (\theta_1, \dots, \theta_M)^T$. The parameters are imprinted onto $\hat{\rho}$ by a unitary evolution $\hat{U}(\boldsymbol{\theta}) = \exp(-i\hat{\mathbf{H}} \cdot \boldsymbol{\theta}) = \exp(-i\sum_{k=1}^{M} \hat{H}_k \theta_k)$, where $\hat{\mathbf{H}} = (\hat{H}_1, \dots, \hat{H}_M)^T$ is a vector of Hamiltonians that do not necessarily commute with each other. After the phase imprinting, a measurement is performed and the experiment is repeated $\mu$ times with the same output state $\hat{\rho}(\boldsymbol{\theta}) = \hat{U}(\boldsymbol{\theta})\hat{\rho}\hat{U}(\boldsymbol{\theta})^\dagger$. The parameters $\theta_k$ are inferred from a set of estimators $\theta_{\mathrm{est},k}$ with $k = 1, \dots, M$, which are functions of the measurement results. The multiparameter uncertainty is quantified by the $M \times M$ covariance matrix $\Sigma$ with elements $\Sigma_{kl} = \mathrm{Cov}(\theta_{\mathrm{est},k}, \theta_{\mathrm{est},l})$. The operational meaning of $\Sigma$ is that, for an arbitrary $M$-dimensional real vector of coefficients $\mathbf{n} = (n_1, \dots, n_M)^T$, the quantity $\mathbf{n}^T\Sigma\mathbf{n} = \Delta^2 (n_1\theta_{\mathrm{est},1} + \cdots + n_M\theta_{\mathrm{est},M})$ yields the variance of the corresponding linear combination of estimators.

We introduce here an estimation protocol based on a multiparameter method of moments. The parameters $\boldsymbol{\theta}$ are estimated from the average values of a set of $K$ measurement observables $\hat{\mathbf{X}} = (\hat{X}_1, \dots, \hat{X}_K)^T$. We consider a commuting set $\hat{\mathbf{X}}$ to ensure simultaneous measurability in a single shot, but our framework does not formally require this assumption. In the central limit, we obtain the covariance matrix (see "Methods" for details).

$$\Sigma = \left(\mu\mathcal{M}[\hat{\rho}(\boldsymbol{\theta}), \hat{\mathbf{H}}, \hat{\mathbf{X}}]\right)^{-1}. \tag{1}$$

The moment matrix,

$$\mathcal{M}[\hat{\rho}(\boldsymbol{\theta}), \hat{\mathbf{H}}, \hat{\mathbf{X}}] = C[\hat{\rho}(\boldsymbol{\theta}), \hat{\mathbf{H}}, \hat{\mathbf{X}}]^T \, \Gamma[\hat{\rho}(\boldsymbol{\theta}), \hat{\mathbf{X}}]^{-1} \, C[\hat{\rho}(\boldsymbol{\theta}), \hat{\mathbf{H}}, \hat{\mathbf{X}}], \tag{2}$$

depends on the covariance matrix $(\Gamma[\hat{\rho}(\boldsymbol{\theta}), \hat{\mathbf{X}}])_{kl} = \langle \hat{X}_k\hat{X}_l \rangle_{\hat{\rho}(\boldsymbol{\theta})} - \langle \hat{X}_k \rangle_{\hat{\rho}(\boldsymbol{\theta})}\langle \hat{X}_l \rangle_{\hat{\rho}(\boldsymbol{\theta})}$ and the commutator matrix $(C[\hat{\rho}(\boldsymbol{\theta}), \hat{\mathbf{H}}, \hat{\mathbf{X}}])_{kl} = -i\langle [\hat{X}_k, \hat{H}_l] \rangle_{\hat{\rho}(\boldsymbol{\theta})}$. Equation (2) provides a lower bound to the classical and quantum Fisher information matrix, i.e.,

$$\mathcal{M}[\hat{\rho}(\boldsymbol{\theta}), \hat{\mathbf{H}}, \hat{\mathbf{X}}] \leq F[\hat{\rho}(\boldsymbol{\theta}), \hat{\mathbf{X}}] \leq F_Q[\hat{\rho}(\boldsymbol{\theta}), \hat{\mathbf{H}}], \tag{3}$$

expressing, e.g., that $F - \mathcal{M}$ is a positive semidefinite matrix[43]. The classical Fisher matrix $F[\hat{\rho}(\boldsymbol{\theta}), \hat{\mathbf{X}}]$ determines the multiparameter sensitivity limit[35,44] attainable by a measurement of the observables $\hat{\mathbf{X}}$ and consists of elements $(F[\hat{\rho}(\boldsymbol{\theta}), \hat{\mathbf{X}}])_{kl} = \sum_{\mathbf{x}} p(\mathbf{x}|\boldsymbol{\theta})\left(\frac{\partial}{\partial\theta_k}\log p(\mathbf{x}|\boldsymbol{\theta})\right)\left(\frac{\partial}{\partial\theta_l}\log p(\mathbf{x}|\boldsymbol{\theta})\right)$, where $p(\mathbf{x}|\boldsymbol{\theta}) = \mathrm{Tr}\{\hat{\Pi}_{\mathbf{x}}\hat{\rho}(\boldsymbol{\theta})\}$ is the probability to obtain the result $\mathbf{x} = (x_1, \dots, x_K)^T$ and the $\hat{\boldsymbol{\Pi}} = \{\hat{\Pi}_{\mathbf{x}}\}_{\mathbf{x}}$ denote the projectors onto the common eigenstates of the $\hat{X}_k$. For any fixed basis, defined by the projectors $\hat{\boldsymbol{\Pi}}$, the bound (Eq. (3)) can be saturated by an optimal choice of the measurement observables $\hat{\mathbf{X}}$ (e.g., by measuring directly the projectors $\hat{\mathbf{X}} = \hat{\boldsymbol{\Pi}}$), leading to

$$\max_{\hat{\mathbf{X}}\in\mathrm{span}(\hat{\boldsymbol{\Pi}})} \mathcal{M}[\hat{\rho}(\boldsymbol{\theta}), \hat{\mathbf{H}}, \hat{\mathbf{X}}] = F[\hat{\rho}(\boldsymbol{\theta}), \hat{\mathbf{X}}]. \tag{4}$$

The short-hand notation $\hat{\mathbf{X}} \in \mathrm{span}(\hat{\boldsymbol{\Pi}})$ expresses that each of the $\hat{X}_k$ is a linear combination of the elements of $\hat{\boldsymbol{\Pi}}$. Moreover, the bound $F[\hat{\rho}(\boldsymbol{\theta}), \hat{\mathbf{X}}] \leq F_Q[\hat{\rho}(\boldsymbol{\theta}), \hat{\mathbf{H}}]$ holds for all $\hat{\mathbf{X}}$, where $(F_Q[\hat{\rho}, \hat{\mathbf{H}}])_{kl} = \mathrm{Tr}\{\hat{\rho}(\hat{L}_k\hat{L}_l + \hat{L}_l\hat{L}_k)/2\}$ is the quantum Fisher information[35,45] and $-i[\hat{H}_k, \hat{\rho}] = (\hat{L}_k\hat{\rho} + \hat{\rho}\hat{L}_k)/2$ defines the symmetric logarithmic derivative operators. The Fisher information matrix equals the quantum Fisher matrix only under certain conditions[36,37]. Equations (3) and (4) and their saturation conditions are derived in Supplementary Note 2. The bounds (Eq. (3)) show that the moment matrix (Eq. (2)) approximates the

state's multiparameter sensitivity by means of first and second moments of the chosen measurement observables $\hat{\mathbf{X}}$. For linear observables $\hat{\mathbf{X}}$ (e.g., collective spins or quadratures), this can be interpreted as a Gaussian approximation of the (quantum) Fisher matrix, but through the measurement of nonlinear observables the method is also able to efficiently characterize non-Gaussian states.

In the following, we present an analytical method for identifying the optimal choice of $\hat{\mathbf{X}}$. We consider here the case of a predefined family of accessible operators $\hat{\mathbf{A}} = (\hat{A}_1, \ldots, \hat{A}_L)^T$ with $L \geq M$ and $K$ that may be chosen as the experimentally available observables. The optimization will be realized under the constraint that only linear combinations of the operators $\hat{\mathbf{A}}$ can be measured. We thus assume that $\hat{\mathbf{X}}$, as well as the Hamiltonians $\hat{\mathbf{H}}$, can be expressed as linear combinations $\hat{H}_k = \sum_{i=1}^L r_{k,i} \hat{A}_i$ and $\hat{X}_k = \sum_{i=1}^L s_{k,i} \hat{A}_i$. The real-valued coefficients $r_{k,i}$ and $s_{k,i}$ define the $M \times L$ and $K \times L$ transformation matrices $R$ and $S$, respectively. We may write

$$\hat{\mathbf{H}} = R\hat{\mathbf{A}}, \quad \text{and} \quad \hat{\mathbf{X}} = S\hat{\mathbf{A}}, \tag{5}$$

and henceforth we assume $RR^T = \mathbb{1}_M$ and $SS^T = \mathbb{1}_K$. We first optimize the choice of the matrix $S$, i.e., the measurement observables, for any fixed phase encoding transformation specified by the matrix $R$. The optimization of the moment matrix (Eq. (2)) is given by

$$\mathcal{M}_{\text{opt}}[\hat{\rho}, \hat{\mathbf{H}}, \hat{\mathbf{A}}] := \max_{\hat{\mathbf{X}} \in \text{span}(\hat{\mathbf{A}})} \mathcal{M}[\hat{\rho}, \hat{\mathbf{H}}, \hat{\mathbf{X}}] = R\tilde{\mathcal{M}}[\hat{\rho}, \hat{\mathbf{A}}] R^T, \tag{6}$$

where

$$\tilde{\mathcal{M}}[\hat{\rho}, \hat{\mathbf{A}}] = \tilde{C}[\hat{\rho}, \hat{\mathbf{A}}]^T \Gamma[\hat{\rho}, \hat{\mathbf{A}}]^{-1} \tilde{C}[\hat{\rho}, \hat{\mathbf{A}}] \tag{7}$$

is the $L \times L$ moment matrix of operators $\hat{\mathbf{A}}$, which is defined on the basis of the covariance matrix $(\Gamma[\hat{\rho}, \hat{\mathbf{A}}])_{kl} = \frac{1}{2} \langle \hat{A}_k \hat{A}_l + \hat{A}_l \hat{A}_k \rangle_{\hat{\rho}} - \langle \hat{A}_k \rangle_{\hat{\rho}} \langle \hat{A}_l \rangle_{\hat{\rho}}$ and the commutator matrix $(\tilde{C}[\hat{\rho}, \hat{\mathbf{A}}])_{kl} = -i \langle [\hat{A}_k, \hat{A}_l] \rangle_{\hat{\rho}}$. The result (Eq. (6)) is proven in Supplementary Note 2 and follows from the matrix inequality

$$\mathcal{M}[\hat{\rho}, \hat{\mathbf{H}}, \hat{\mathbf{X}}] \leq R\tilde{\mathcal{M}}[\hat{\rho}, \hat{\mathbf{A}}]R^T, \tag{8}$$

which holds for arbitrary $\hat{\mathbf{X}}$. Saturation in Eq. (8) is achieved by the observables defined in Eq. (5) if and only if there exists a real-valued $K \times M$ matrix $G$ such that

$$GS = R\tilde{C}[\hat{\rho}, \hat{\mathbf{A}}]^T \Gamma[\hat{\rho}, \hat{\mathbf{A}}]^{-1}. \tag{9}$$

This result generalizes the analytical optimization discussed in ref. [46] to the multiparameter case. Moreover, the choice of parameter-encoding Hamiltonians, i.e., $R$ can be optimized by considering the spectrum of $\tilde{\mathcal{M}}[\hat{\rho}, \hat{\mathbf{A}}]$ (see "Methods"). In practice, the optimal moment matrix (Eq. (6)) can only be achieved by a direct measurement if the elements of an optimal $\hat{\mathbf{X}}$, defined by Eq. (9), can be measured simultaneously.

**Squeezing matrix.** We define the squeezing matrix by comparing the moment-based sensitivity $\Sigma$ of Eq. (1) to the multiparameter shot-noise limit $\Sigma_{\text{SN}}$, i.e., the sensitivity limit of classical measurement strategies. While this approach can be applied to arbitrary multiparameter estimation scenarios, in the following we focus mostly on the experimentally relevant cases of distributed sensor networks or multimode interferometers[15–17]: The parameters are encoded in $M$ different modes by local Hamiltonians satisfying $[\hat{H}_k, \hat{H}_l] = 0$ for all $k$, $l$ and we measure one observable in each mode ($K = M$). In these cases, the shot-noise limit $\Sigma_{\text{SN}} =$

$(\mu F_{\text{SN}}[\hat{\mathbf{H}}])^{-1}$ can be explicitly determined from the quantum Cramér–Rao bound (see "Methods"). For evolutions generated by $\hat{\mathbf{H}}$ and measurement observables $\hat{\mathbf{X}}$, we define the squeezing matrix as

$$\Xi^2[\hat{\rho}, \hat{\mathbf{H}}, \hat{\mathbf{X}}] := F_{\text{SN}}[\hat{\mathbf{H}}]^{\frac{1}{2}} \mathcal{M}[\hat{\rho}, \hat{\mathbf{H}}, \hat{\mathbf{X}}]^{-1} F_{\text{SN}}[\hat{\mathbf{H}}]^{\frac{1}{2}}. \tag{10}$$

By expressing Eq. (1) as $\Sigma = \Sigma_{\text{SN}}^{\frac{1}{2}} \Xi^2[\hat{\rho}, \hat{\mathbf{H}}, \hat{\mathbf{X}}] \Sigma_{\text{SN}}^{\frac{1}{2}}$, we observe that the squeezing matrix $\Xi^2[\hat{\rho}, \hat{\mathbf{H}}, \hat{\mathbf{X}}]$ directly quantifies the quantum gain in a saturable, moment-based multiparameter estimation protocol. Any quantum state with the property

$$\Xi^2[\hat{\rho}, \hat{\mathbf{H}}, \hat{\mathbf{X}}] \geq \mathbb{1}_M \tag{11}$$

can only yield multiparameter shot-noise sensitivity or worse, i.e., $\Sigma \geq \Sigma_{\text{SN}}$. Inserting Eq. (6) into Eq. (10), we obtain the optimized squeezing matrix:

$$\Xi^2_{\text{opt}}[\hat{\rho}, \hat{\mathbf{H}}, \hat{\mathbf{A}}] := \min_{\hat{\mathbf{X}} \in \text{span}(\hat{\mathbf{A}})} \Xi^2[\hat{\rho}, \hat{\mathbf{H}}, \hat{\mathbf{X}}]$$
$$= F_{\text{SN}}[\hat{\mathbf{H}}]^{\frac{1}{2}} R\tilde{\mathcal{M}}[\hat{\rho}, \hat{\mathbf{A}}]^{-1} R^T F_{\text{SN}}[\hat{\mathbf{H}}]^{\frac{1}{2}}. \tag{12}$$

A violation of the matrix inequality (Eq. (11)) signals multiparameter squeezing (with respect to the phase-imprinting Hamiltonians $\hat{\mathbf{H}}$ and the measurement observables $\hat{\mathbf{X}}$): it implies that there exists at least one vector $\mathbf{n} \in \mathbb{R}^M$ for which $\mathbf{n}^T \Sigma \mathbf{n} < \mathbf{n}^T \Sigma_{\text{SN}} \mathbf{n}$ holds. In this case, sub-shot-noise sensitivity is achieved for the estimation of $\mathbf{n}^T \boldsymbol{\theta}$, which describes a particular linear combination of the parameters. The number $0 \leq r_{\text{SN}} \leq M$ of negative eigenvalues of the matrix $\Sigma - \Sigma_{\text{SN}}$ defines the shot-noise rank[17] that is achieved by the multiparameter method of moments. Equivalently, $r_{\text{SN}}$ corresponds to the number of eigenvalues of $\Xi^2$ that are smaller than one. When $r_{\text{SN}} = M$, the stronger condition $\Xi^2[\hat{\rho}, \hat{\mathbf{H}}, \hat{\mathbf{X}}] < \mathbb{1}_M$ for full multiparameter squeezing is satisfied. In this case, $\Sigma < \Sigma_{\text{SN}}$ holds, and sub-shot-noise sensitivity is achieved for the estimation of arbitrary $\mathbf{n}^T \boldsymbol{\theta}$.

The observation of multiparameter squeezing implies that the state is nonclassical (see "Methods"). To increase the quantum enhancements, it is thus beneficial to reduce the squeezing matrix as much as possible by using nonclassical states.

**Multiparameter discrete-variable (spin) squeezing.** Discrete-variable multiparameter estimation provides the theoretical framework to model a series of $M$ local Ramsey or Mach–Zehnder interferometers that operate in parallel, each with a fixed number of particles $N_k$, with $k = 1, \ldots, M$; see Fig. 1. Here each mode is modeled by a collective spin of length $N_k/2$, for $k = 1, \ldots, M$, summing up to a total number of $N = \sum_{k=1}^M N_k$ spin-1/2 particles.

The multimode interferometer is described by a family of local parameter-encoding Hamiltonians $\hat{\mathbf{H}} = \hat{\mathbf{J}}_{\mathbf{r}} = (\hat{J}_{\mathbf{r}_1,1}, \ldots, \hat{J}_{\mathbf{r}_M,M})^T$, where $\mathbf{r} = (\mathbf{r}_1, \ldots, \mathbf{r}_M)$, $\hat{J}_{\mathbf{r}_k,k} = \mathbf{r}_k^T \hat{\mathbf{J}}_{\perp,k}$, $\hat{\mathbf{J}}_{\perp,k} = (\hat{J}_{x,k}, \hat{J}_{y,k})^T$, and $\hat{J}_{\alpha,k} = \sum_{i=1}^{N_k} \hat{\sigma}_{\alpha,k}^{(i)}/2$ is a collective spin operator on mode $k$ with Pauli matrices $\hat{\sigma}_{\alpha,k}^{(i)}$ for $\alpha = x, y, z$ and $k = 1, \ldots, M$. Without loss of generality, we label the axes such that the mean spin direction $\mathbf{n}_{0,k} = \langle \hat{\mathbf{J}}_k \rangle_{\hat{\rho}} / |\langle \hat{\mathbf{J}}_k \rangle_{\hat{\rho}}|$ defines the $z$ axis. By considering a family of local measurement observables $\hat{\mathbf{X}} = \hat{\mathbf{J}}_{\mathbf{s}}$, we obtain the spin-squeezing matrix with elements

$$\left(\Xi^2[\hat{\rho}, \hat{\mathbf{J}}_{\mathbf{r}}, \hat{\mathbf{J}}_{\mathbf{s}}]\right)_{kl} = \frac{\sqrt{N_k N_l}\,\text{Cov}\left(\hat{J}_{\mathbf{s}_k,k}, \hat{J}_{\mathbf{s}_l,l}\right)_{\hat{\rho}}}{\langle \hat{J}_{z,k} \rangle_{\hat{\rho}} \langle \hat{J}_{z,l} \rangle_{\hat{\rho}}}, \tag{13}$$

where we used Eq. (10) with $F_{\text{SN}}[\hat{\mathbf{J}}_{\mathbf{r}}]^{\frac{1}{2}} C[\hat{\rho}, \hat{\mathbf{J}}_{\mathbf{r}}, \hat{\mathbf{J}}_{\mathbf{s}}]^{-1} =$

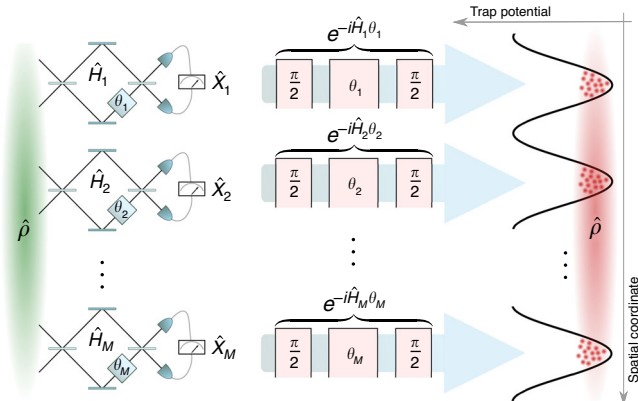

**Fig. 1 Quantum-enhanced parallel interferometers.** In each mode $k = 1, ..., M$ of a set of Mach–Zehnder (left) or Ramsey interferometers (right), a single parameter $\theta_k$ is imprinted by a local Hamiltonian $\hat{H}_k$, and a local observable $\hat{X}_k$ is measured. The multiparameter sensitivity is quantified by the moment matrix (Eq. (2)). The multiparameter quantum gain is captured by the squeezing matrix (Eq. (10)), which contains both local (single-parameter) enhancements and nonlocal (multiparameter) squeezing. The sensitivity can be optimized analytically using Eq. (6) and the maximum is achieved when Eq. (9) is fulfilled for a set of commuting observables $\hat{X}_1, ..., \hat{X}_M$.

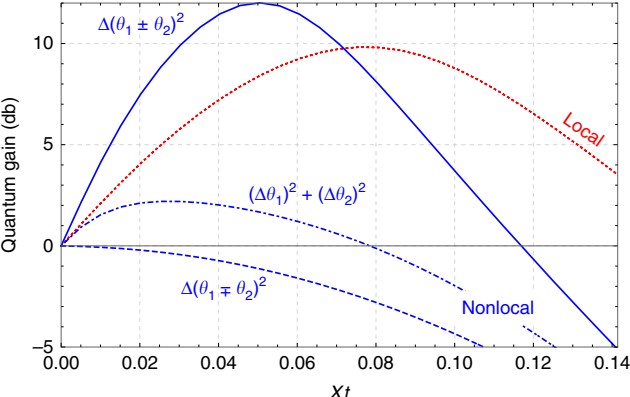

**Fig. 2 Local vs nonlocal atomic spin squeezing.** For a local parameter encoding with $N = 100$ particles, nonlocal squeezing, described by Eq. (14), leads to a larger quantum sensitivity gain for either the sum $10\log_{10}(\mathbf{n}_+^T \Sigma_{\mathrm{SN}} \mathbf{n}_+ / \mathbf{n}_+^T \Sigma \mathbf{n}_+)$ (continuous blue line) or the difference of two spatially distributed parameters $10\log_{10}(\mathbf{n}_-^T \Sigma_{\mathrm{SN}} \mathbf{n}_- / \mathbf{n}_-^T \Sigma \mathbf{n}_-)$ (dashed blue line) than local squeezing, Eq. (15). Since the spin-squeezing matrix is diagonal when squeezing is local, both combinations of parameters, as well as their uncorrelated average, yield the same sensitivity (red dashed line). Nonlocal squeezing yields a lower quantum gain for the uncorrelated average $10\log_{10}(\mathrm{Tr}\Sigma_{\mathrm{SN}}/\mathrm{Tr}\Sigma)$ (dashed-dotted line). The plot shows data for local directions $\mathbf{r}_1$ and $\mathbf{r}_2$ chosen to maximize the gain for the sum. A local rotation transforms the sum of parameters into the difference and vice versa.

$\mathrm{diag}(\sqrt{N_1}/\langle \hat{J}_{z,1}\rangle_{\hat{\rho}}, ..., \sqrt{N_M}/\langle \hat{J}_{z,M}\rangle_{\hat{\rho}})$ and we assumed that the $\mathbf{r}_k$ and $\mathbf{s}_k$ are orthonormal vectors in the $xy$-plane, such that $\langle \hat{J}_{z,k}\rangle_{\hat{\rho}} = -i\langle[\hat{J}_{\mathbf{s}_k,k}, \hat{J}_{\mathbf{r}_k,k}]\rangle_{\hat{\rho}}$ is the length of spin $k$ with mean spin direction along the $z$ axis. On its diagonal, this matrix contains the local spin-squeezing coefficients[5] for each of the modes $k = 1, ..., M$. It is well known that these coefficients reveal the number of entangled spins within the local modes[1,7,8]. In addition to these single-parameter contributions, the multiparameter spin-squeezing matrix (Eq. (13)) includes off-diagonal terms that are due to mode correlations, i.e., entanglement between the individual interferometers.

**Atomic multiparameter spin squeezing.** A locally squeezed state can be created by subjecting spatially separated ensembles of atoms to local, nonlinear evolutions, e.g., by means of the one-axis twisting Hamiltonian[6]. It is easy to see from the squeezing matrix that local squeezing is sufficient to attain full multiparameter sub-shot noise; see Supplementary Note 3 for details. However, atomic experiments are not limited to the generation of local squeezing: recently, spatially distributed entanglement was observed by splitting squeezed atomic spin ensembles into two or more external modes[47–49].

In order to identify the metrological potential of nonlocal squeezing, we compare two different spin-squeezing strategies. We consider an even number of $N$ spin-1/2 particles initialized in the polarized state $|\Psi_0\rangle = |\uparrow\rangle^{\otimes N}$, where $|\uparrow\rangle$ is an eigenstate of the Pauli $z$ matrix. Local squeezing (namely, local in each atomic ensemble) corresponds to

$$|\Psi_{\mathrm{loc}}(t)\rangle = e^{-i(\hat{J}_{y,1}^2 + \hat{J}_{y,2}^2)\chi t}|\Psi_0\rangle, \tag{14}$$

where $\hat{J}_{y,1}$ and $\hat{J}_{y,2}$ are collective spin operators for particles 1, 2, ..., $N/2$ and $N/2 + 1$, ..., $N$, respectively, i.e., we have separated the particles into two ensembles of equal size. The nonlinear evolution generates entanglement between the $N/2$ particles in each ensemble, e.g., by describing interactions among the particles in the same ensemble for the dimensionless time $\chi t$ but does not entangle the two ensembles. Nonlocal squeezing is

instead described by the collective one-axis-twisting evolution

$$|\Psi_{\mathrm{nl}}(t)\rangle = e^{-i(\hat{J}_{y,1} + \hat{J}_{y,2})^2 \chi t}|\Psi_0\rangle, \tag{15}$$

which creates particle entanglement between the $N$ spins and mode entanglement between the two ensembles.

Our goal is to estimate linear combinations $\mathbf{n}^T\boldsymbol{\theta} = n_1\theta_1 + n_2\theta_2$ of locally encoded parameters, generated by the rotations $\hat{J}_{\mathbf{r}_1,1}$ and $\hat{J}_{\mathbf{r}_2,2}$ via the transformation $\hat{U}(\boldsymbol{\theta}) = \exp(-i\hat{J}_{\mathbf{r}_1,1}\theta_1 - i\hat{J}_{\mathbf{r}_2,2}\theta_2)$. A particular case of interest is the estimation of a magnetic field gradient[30–33] based on the differential measurement of the field at two spatially separated locations, which corresponds to the difference $\mathbf{n}_- = (1, -1)^T/\sqrt{2}$. A related task is the estimation of the average field, i.e., the sum of parameters $\mathbf{n}_+ = (1, 1)^T/\sqrt{2}$. We assume that the local rotation axes $\mathbf{r}_1$ and $\mathbf{r}_2$ (and their corresponding optimal measurement directions) can be adjusted to optimize the local squeezing parameters, i.e., to minimize the diagonal entries of the squeezing matrix (Eq. (13)). Such a change of the rotation axis can effectively be realized through local rotations of the respective spin states before the interferometric measurement[1].

The resulting sensitivities for $\mathbf{n}_\pm^T\boldsymbol{\theta}$ are compared in Fig. 2 for an ensemble of $N = 100$ atoms as a function of the nonlinear evolution time $t$. We observe that an estimation of, e.g., $\mathbf{n}_+^T\boldsymbol{\theta}$ can be enhanced by nonlocal squeezing (blue continuous line). As a consequence, the sensitivity for $\mathbf{n}_+^T\boldsymbol{\theta}$ is reduced below the classical limit (blue dashed line). However, a local $\pi$-rotation of the state can effectively change the sign of $\mathbf{r}_2$ and transform the sum into the difference and vice-versa. Hence, nonlocal squeezing can be used to reduce the uncertainty of a specific linear combination of parameters. The state cannot be optimal for arbitrary linear combinations at the same time, but local operations can be used to adjust the state prior to the measurement in order to optimally harness the nonlocal squeezing and beat the sensitivity of local squeezing. Nonlocal squeezing further improves the estimation of

nonlocally encoded parameters, as we discuss in Supplementary Note 3.

**Multiparameter continuous-variable squeezing.** Continuous-variable multiparameter estimation studies the sensitivity to a multimode displacement described by phase space operators $\hat{\mathbf{q}} = (\hat{x}_1, \hat{p}_1, \ldots, \hat{x}_M, \hat{p}_M)^T$, where $\hat{x}_k = \frac{1}{2}(\hat{a}_k + \hat{a}_k^\dagger)$ and $\hat{p}_k = \frac{1}{2i}(\hat{a}_k - \hat{a}_k^\dagger)$ and $[\hat{a}_k, \hat{a}_{k'}^\dagger] = \delta_{kk'}$. These observables are accessible by homodyne measurement techniques, i.e., by mixing the signal with a strongly populated local oscillator—a well-established technique in optical[10–13,50,51] and atomic systems[52,53].

The $2M \times 2M$ moment matrix, Eq. (7), for $\hat{\mathbf{A}} = \hat{\mathbf{q}}$ reads

$$\tilde{\mathcal{M}}[\hat{\rho}, \hat{\mathbf{q}}] = \frac{1}{4}\Omega^T \Gamma[\hat{\rho}, \hat{\mathbf{q}}]^{-1} \Omega \tag{16}$$

and provides the maximally achievable sensitivity for multimode displacements via Eq. (6). The $2M \times 2M$ covariance matrix $\Gamma[\hat{\rho}, \hat{\mathbf{q}}]$ contains complete information on non-displaced Gaussian states. The commutator matrix $\tilde{C}[\hat{\rho}, \hat{\mathbf{q}}] = \frac{1}{2}\Omega$ is independent of the quantum state, where $\Omega = \bigoplus_{k=1}^{M} \omega$ is the symplectic form with $\omega = \begin{pmatrix} 0 & 1 \\ -1 & 0 \end{pmatrix}$[11–13]. Furthermore, the explicit evaluation of the quantum Fisher matrix of Gaussian states $\hat{\rho}_G$[54–56] (i.e., states whose Wigner function is Gaussian[11–13]) reveals that it coincides with Eq. (16). We thus obtain the exact equality $\tilde{\mathcal{M}}[\hat{\rho}_G, \hat{\mathbf{q}}] = F_Q[\hat{\rho}_G, \hat{\mathbf{q}}]$ for arbitrary Gaussian states $\hat{\rho}_G$, whereas for arbitrary quantum states $\hat{\rho}$, Eq. (16) represents a Gaussian lower bound to the quantum Fisher matrix, see Eq. (3). Making use of upper bounds on the quantum Fisher matrix for specific classes of separable states[17,57,58], the moment matrix can reveal detailed information about the multimode entanglement structure[59].

The continuous-variable squeezing matrix, optimized over the measurement observables $\hat{\mathbf{X}}$, is given by Eq. (12) and reads:

$$\Xi^2_{\text{opt}}[\hat{\rho}, \hat{\mathbf{H}}, \hat{\mathbf{q}}] = 4R\Omega^T \Gamma[\hat{\rho}, \hat{\mathbf{q}}]\Omega R^T. \tag{17}$$

Let us first revisit the general squeezing condition for the particular case of the multimode continuous-variable system at hand. A violation of Eq. (11) implies that (see Supplementary Note 4)

$$\lambda_{\min}(\Gamma[\hat{\rho}, \hat{\mathbf{q}}]) < \frac{1}{4}, \tag{18}$$

where $\lambda_{\min}$ denotes the smallest eigenvalue. The condition Eq. (18) was originally proposed in ref. [60] as a definition of squeezing in multimode continuous-variable systems that is invariant under passive transformations, i.e., beam splitter operations and phase shifters that leave the number of photons constant. Conversely, if Eq. (18) holds, one can find $\hat{\mathbf{H}}$ and $\hat{\mathbf{X}}$ such that the condition Eq. (11) is violated.

Hence, our general metrological definition of squeezing in multimode systems is equivalent to a well-established definition[60] in the continuous-variable case when considering quadrature operators. The shot-noise rank $r_{\text{SN}}$, i.e., the number of eigenvalues of $\Xi^2_{\text{opt}}[\hat{\rho}, \hat{\mathbf{H}}, \hat{\mathbf{q}}]$ that are smaller than one, provides a step-wise characterization of the multiparameter quantum gain up to full multiparameter squeezing (namely, $r_{\text{SN}} = M$). This establishes a natural multiparameter extension of the single-parameter condition (Eq. (18)), which merely implies that $r_{\text{SN}} > 0$.

**Multimode squeezed vacuum states.** The class of pure Gaussian continuous-variable states is given by multimode squeezed vacuum states $|\Psi_0\rangle$[10–13]. As a consequence of the Williamson theorem and the Bloch–Messiah decomposition[61], any such state can be generated by a combination of local squeezing and a series of passive operations[11,13]. Consequently, there always exists a

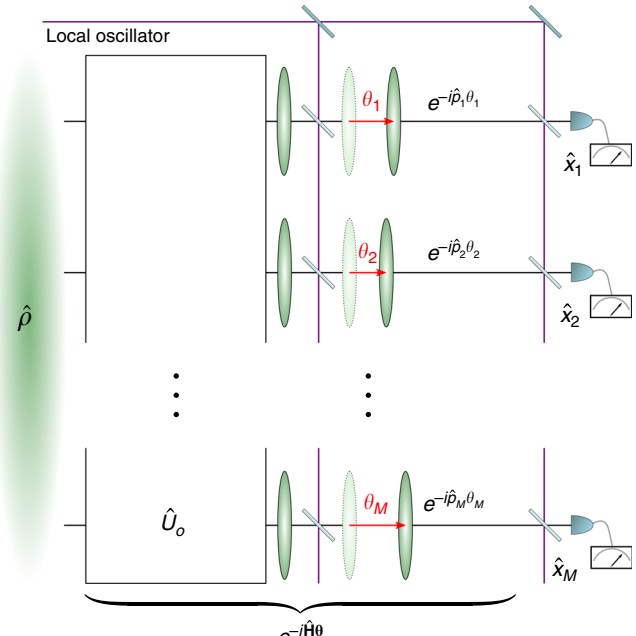

**Fig. 3 Optimal multimode displacement sensing with squeezed vacuum states.** The passive transformation $\hat{U}_O$ decouples the initial multimode squeezed vacuum state into local squeezed states. A displacement generated by the anti-squeezed variance (here depicted as $\hat{p}$) and a measurement of the squeezed variance ($\hat{x}$) is implemented in each mode with the aid of a local oscillator.

$2M \times 2M$ orthogonal symplectic matrix $O$ and a corresponding passive operation described by $\hat{U}_O$ that yields $\Gamma[\hat{U}_O|\Psi_0\rangle, \hat{\mathbf{q}}] = O\Gamma[|\Psi_0\rangle, \hat{\mathbf{q}}]O^T = \frac{1}{4}\bigoplus_{k=1}^{M} \text{diag}(e^{2r_k}, e^{-2r_k})$, where $r_1, \ldots, r_M$ quantify the squeezing in each of the modes.

The choice of phase-encoding Hamiltonians and measurement observables $\hat{\mathbf{H}} = UP_M O\Omega\hat{\mathbf{q}}$ and $\hat{\mathbf{X}} = P_M O\hat{\mathbf{q}}$, where $U$ is an arbitrary $M \times M$ orthogonal matrix and $P_M$ is a $M \times 2M$ projector that picks one quadrature per mode, is optimal (see "Methods" and Supplementary Note 4 for details) and leads to

$$\Xi^2_{\text{opt}}[|\Psi_0\rangle, \hat{\mathbf{H}}, \hat{\mathbf{q}}] = U\begin{pmatrix} e^{-2r_1} & \cdots & 0 \\ \vdots & \ddots & \vdots \\ 0 & \cdots & e^{-2r_M} \end{pmatrix} U^T. \tag{19}$$

These operators can be interpreted (see Fig. 3) as a phase-imprinting evolution that first disentangles the state and then implements local phase shifts along the respective squeezed quadrature in each mode. The measurement is realized in the corresponding conjugate quadrature. If all $r_k > 0$, we have full multiparameter squeezing, enabling sub-shot-noise estimation of arbitrary linear combinations of the parameters $\boldsymbol{\theta}$ encoded via the evolution $U(\boldsymbol{\theta}) = e^{-i(\hat{H}_1\theta_1 + \cdots + \hat{H}_M\theta_M)}$.

For $O = 1_{2M}$ and $U = 1_M$, the parameter encoding realized by the Hamiltonians $\hat{\mathbf{H}}$ is local in the modes $\hat{\mathbf{q}}$. The result (Eq. (19)) shows that the multiparameter sensitivity of local transformations is maximized by a mode-local product state. Similarly, for any other choice of $O$, we can define new modes $O\hat{\mathbf{q}}$ as nonlocal linear combinations of the original $\hat{\mathbf{q}}$, and for transformations that are local in $O\hat{\mathbf{q}}$, the sensitivity is maximized by states that are uncorrelated in the modes $O\hat{\mathbf{q}}$. These states will generally be mode entangled in the original set of modes $\hat{\mathbf{q}}$. We conclude that mode entanglement with respect to the modes $\hat{\mathbf{q}}$ is not necessary to optimize the overall multiparameter sensitivity if the parameter encoding is done locally in $\hat{\mathbf{q}}$. Conversely, given a transformation

that is nonlocal in $\hat{\mathbf{q}}$, the optimal sensitivity is achieved by a mode entangled state.

**Maximum enhancement due to mode entanglement.** Recall that the multiparameter covariance matrix contains information equivalent to the sensitivity of arbitrary linear combinations of parameters. For any specific linear combination, local squeezing is still suboptimal (an analog observation was discussed above for the case of spins). In this case, we are interested in minimizing a single matrix element rather than all eigenvalues of the squeezing matrix. Let us now identify the maximum gain that can be achieved by making use of mode entanglement.

We consider a fixed family of phase-imprinting Hamiltonians (hence $U = 1_M$) and an estimation of $\mathbf{n}^T\boldsymbol{\theta}$ with an arbitrary, fixed unit vector $\mathbf{n}$ that has non-zero overlap with all the participating modes $k = 1, \ldots, M$. Our goal is to distribute a finite total amount of squeezing (determined by the total average particle number) over all modes in order to minimize $\mu\mathbf{n}^T\Sigma\mathbf{n} = \mathbf{n}^T\Xi^2_{\mathrm{opt}}[|\Psi_0\rangle, \hat{\mathbf{H}}, \mathbf{q}]\mathbf{n}$. We compare the optimized mode-separable and mode-entangled strategy (see "Methods" for details), giving rise to the respective sensitivities $(\Delta\theta_{\mathrm{m-sep}})^2$ and $(\Delta\theta_{\mathrm{m-ent}})^2$: For a uniform average over all parameters, $n_k = 1/\sqrt{M}$, the optimal mode-separable strategy consists in equal squeezing in all modes, $r_k = r$, for $k = 1, \ldots, M$, while the optimal mode-entangled strategy concentrates all squeezing into a single mode. As soon as $r > 0$, we have $(\Delta\theta_{\mathrm{m-ent}})^2/(\Delta\theta_{\mathrm{m-sep}})^2 < 1$, see Fig. 4: the mode-entangled strategy outperforms the mode-separable one. In the limit $r \ll 1/\sqrt{M}$, we obtain $(\Delta\theta_{\mathrm{m-ent}})^2/(\Delta\theta_{\mathrm{m-sep}})^2 \approx e^{-2(\sqrt{M}-1)r}$. In the opposite limit, $r \gg 1$, we have $e^{-2r'} \approx Me^{-2r}$ and we obtain

$$\frac{(\Delta\theta_{\mathrm{m-ent}})^2}{(\Delta\theta_{\mathrm{m-sep}})^2} = \frac{1}{M} \qquad (r \gg 1). \tag{20}$$

We thus recover the gain factor $1/M$ that has been identified as the maximal gain due to mode entanglement[14–17,23]. Here the factor $1/M$ is obtained by comparing optimal Gaussian states based on the analysis of the multimode squeezing matrix. We further show in Supplementary Note 4 that, among all possible

states with fixed average particle number, squeezed vacuum states optimize the sensitivity of multiparameter displacement sensing, generalizing the single-parameter results of refs. [62,63].

**Non-commuting generators and non-Gaussian states.** To illustrate how our methods can lead to efficient and saturable strategies in more general scenarios, we now discuss an example dedicated to the estimation of parameters that are generated by non-commuting operators using a non-Gaussian state.

We consider the estimation of the two angles $\theta_{1,2}$ of a SU(2) rotation $\hat{U}(\boldsymbol{\theta}) = e^{-i(\theta_1\hat{J}_x + \theta_2\hat{J}_y)}$ with non-commuting generators $\hat{\mathbf{H}} = (\hat{J}_x, \hat{J}_y)^T$ in a single mode. As probe, we use the twin-Fock state $|\mathrm{TF}\rangle$, i.e., eigenstates of $\hat{J}_z$ with eigenvalue zero and a total spin length of $N/2$, and we denote $|\mathrm{TF}_{\boldsymbol{\theta}}\rangle = \hat{U}(\boldsymbol{\theta})|\mathrm{TF}\rangle$. Having zero mean spin length, $|\mathrm{TF}\rangle$ cannot be characterized by spin squeezing[64] and Gaussian measurements are unable to fully harness its metrological potential. We consider the two commuting nonlinear observables $\hat{X}_1 = \hat{J}_x|\mathrm{TF}\rangle\langle\mathrm{TF}|\hat{J}_x$ and $\hat{X}_2 = \hat{J}_y|\mathrm{TF}\rangle\langle\mathrm{TF}|\hat{J}_y$: as a consequence of $\langle\mathrm{TF}|\hat{J}_x\hat{J}_y|\mathrm{TF}\rangle = 0$, we have $[\hat{X}_1, \hat{X}_2] = 0$. Let us indicate with $Q = \langle\mathrm{TF}|\hat{J}_x^2|\mathrm{TF}\rangle = \langle\mathrm{TF}|\hat{J}_y^2|\mathrm{TF}\rangle = \frac{N(N+2)}{8}$. To the leading order in $\theta_{1,2}$, we obtain the inverse covariance matrix $\Gamma[|\mathrm{TF}_{\boldsymbol{\theta}}\rangle, \hat{\mathbf{X}}]^{-1} = Q^{-3}\mathrm{diag}(\theta_1^{-2}, \theta_2^{-2})$ and the commutator matrix $C[|\mathrm{TF}_{\boldsymbol{\theta}}\rangle, \hat{\mathbf{H}}, \hat{\mathbf{X}}] = 2Q^2\mathrm{diag}(\theta_1, \theta_2)$. In the limit $\boldsymbol{\theta} \to 0$, this leads to the moment matrix (Eq. (2))

$$\mathcal{M}[|\mathrm{TF}\rangle, \hat{\mathbf{H}}, \hat{\mathbf{X}}] = \frac{N(N+2)}{2}1_2, \tag{21}$$

which coincides with the quantum Fisher matrix $F_Q[|\mathrm{TF}\rangle, \hat{\mathbf{H}}]$. This shows that, through the measurement of nonlinear observables, our method can extract the full sensitivity of non-Gaussian states and that it can achieve the ultimate multiparameter sensitivity limit even when the generators do not commute.

## Discussion

We introduced metrological multiparameter squeezing as a practical framework to characterize the sensitivity and quantum gain of multiparameter estimation. Our optimization technique can be adapted to any set of accessible observables and thereby allows to adjust the level of complexity to the problem at hand. For example, the multiparameter sensitivity of Gaussian states can be fully captured by a squeezing matrix only containing first and second moments of linear observables. The analysis of the squeezing matrix reveals optimal strategies for the design and analysis of atomic and photonic experiments where Gaussian states still represent the best-controlled and most efficiently generated class of states for metrology. Metrological multiparameter squeezing thus lays the foundation for the development of atomic clocks and electromagnetic field sensors, enhanced by non-local quantum correlations in atomic ensembles with spatially distributed and accessible entanglement[47–49,65–69]. Furthermore, optical systems provide an established platform with access to entangled multimode photonic quantum states[50,51,70] that can be combined with squeezing[71,72]. Our theory of multiparameter squeezing provides a common framework to characterize these experiments and to interpret and optimize them for multiparameter quantum-sensing applications.

By extending the set of accessible observables, the squeezing matrix can be generalized to yield more powerful quantifiers of multiparameter sensitivity that are able to cope with highly sensitive features of non-Gaussian multimode states. This method can also be applied in non-commuting scenarios, where, however,

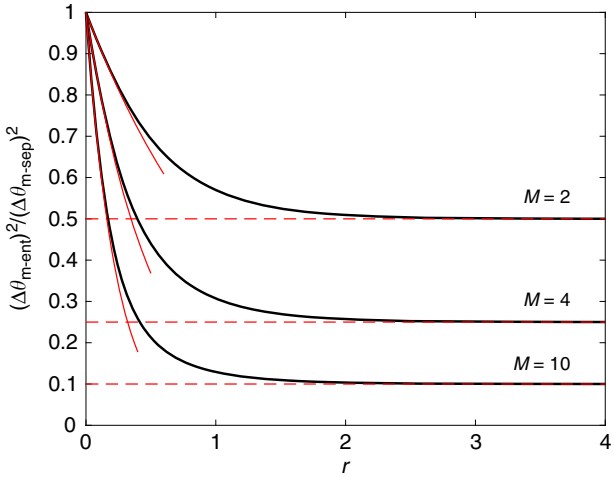

**Fig. 4 Quantum gain from nonlocal mode entanglement.** We plot the ratio between the sensitivity to an uniform average of parameters $(\Delta\theta)^2 = \mathbf{n}^T\Sigma\mathbf{n}$ for optimal mode-entangled and mode-separable states (thick black line), as a function of the squeezing parameter $r$. The solid red lines are the small-$r$ approximation $e^{-2(\sqrt{M}-1)r}$ and the dashed red lines are the large-$r$ approximation $1/M$. Different sets of lines refer to different values of $M$.

further studies are needed to explore the full potential of our approach. Such developments are important, e.g., in optical systems where one aims to estimate the coordinates of an ensemble of emitters to reconstruct an image[24–28,71]. The identification of fundamental resolution limits for quantum imaging requires experimentally and theoretically accessible measures of multiparameter sensitivity for arbitrary emitters.

## Methods

**Multiparameter method of moments.** We base our multiparameter method of moments on the knowledge of the mean values of a family of commuting observables, $\langle \hat{\mathbf{X}} \rangle_{\hat{\rho}(\boldsymbol{\theta})} = (\langle \hat{X}_1 \rangle_{\hat{\rho}(\boldsymbol{\theta})}, \dots, \langle \hat{X}_K \rangle_{\hat{\rho}(\boldsymbol{\theta})})^T$, obtained from the calibration of the experimental apparatus as a function of the $M$ parameters $\boldsymbol{\theta} = (\theta_1, \dots, \theta_M)^T$. If $\hat{\mathbf{X}}$ is measured $\mu \gg 1$ times, each of its components $\hat{X}_k$ yields a sequence of results $x_k^{(1)}, \dots, x_k^{(\mu)}$ where the $x_k^{(i)}$ are picked from the eigenvalues of $\hat{X}_k$. Each measurement of $\hat{\mathbf{X}}$ thus yields a vector of results $\mathbf{x}^{(i)} = (x_1^{(i)}, \dots, x_K^{(i)})^T$ that is randomly distributed with mean value $\langle \hat{\mathbf{X}} \rangle_{\hat{\rho}(\boldsymbol{\theta})}$ and covariance matrix $(\Gamma[\hat{\rho}(\boldsymbol{\theta}), \hat{\mathbf{X}}])_{kl} = \langle \hat{X}_k \hat{X}_l \rangle_{\hat{\rho}(\boldsymbol{\theta})} - \langle \hat{X}_k \rangle_{\hat{\rho}(\boldsymbol{\theta})} \langle \hat{X}_l \rangle_{\hat{\rho}(\boldsymbol{\theta})}$. From these measurements, we obtain the sample average $\bar{\mathbf{X}}^{(\mu)} = (\bar{X}_1^{(\mu)}, \dots, \bar{X}_K^{(\mu)})^T$ with $\bar{X}_k^{(\mu)} = \frac{1}{\mu} \sum_{i=1}^{\mu} x_k^{(i)}$ for $k = 1, \dots, K$. We estimate the parameters $\boldsymbol{\theta}$ as the values for which $\langle \hat{X}_k \rangle_{\hat{\rho}(\boldsymbol{\theta})} = \bar{X}_k^{(\mu)}$ holds for all $k = 1, \dots, K$. As a consequence of the multivariate central limit theorem (see Supplementary Note 1 for details), for $\mu \gg 1$, this strategy yields $\Sigma = (\mu \mathcal{M}[\hat{\rho}(\boldsymbol{\theta}), \hat{\mathbf{X}}])^{-1}$, where

$$\mathcal{M}[\hat{\rho}(\boldsymbol{\theta}), \hat{\mathbf{X}}] = D[\hat{\rho}(\boldsymbol{\theta}), \hat{\mathbf{X}}]^T \Gamma[\hat{\rho}(\boldsymbol{\theta}), \hat{\mathbf{X}}]^{-1} D[\hat{\rho}(\boldsymbol{\theta}), \hat{\mathbf{X}}] \quad (22)$$

and

$$\left( D[\hat{\rho}(\boldsymbol{\theta}), \hat{\mathbf{X}}] \right)_{kl} = \frac{\partial \langle \hat{X}_k \rangle_{\hat{\rho}(\boldsymbol{\theta})}}{\partial \theta_l}. \quad (23)$$

In the case of a single parameter estimated by a single observable ($M = K = 1$), we obtain a sensitivity described by the familiar error propagation formula $(\Delta \theta_{\text{est}})^2 = \frac{1}{\mu} (\Delta \hat{H})^2_{\hat{\rho}(\theta)} \left| \frac{\partial \langle \hat{X} \rangle_{\hat{\rho}(\theta)}}{\partial \theta} \right|^{-2}$, where $\frac{\partial \langle \hat{X} \rangle_{\hat{\rho}(\theta)}}{\partial \theta} = -i \langle [\hat{X}, \hat{H}] \rangle_{\hat{\rho}(\theta)}$[5]. The result (Eq. (22)) provides a direct generalization to the multiparameter case.

For a unitary phase imprinting processes $U(\boldsymbol{\theta}) = \exp(-i \hat{\mathbf{H}} \boldsymbol{\theta})$, generated by the vector of Hamiltonians $\hat{\mathbf{H}} = (\hat{H}_1, \dots, \hat{H}_M)^T$, as considered in the main text, we obtain

$$\left( D[\hat{\rho}(\boldsymbol{\theta}), \hat{\mathbf{X}}] \right)_{kl} = -i \langle [\hat{X}_k, \hat{H}_l] \rangle_{\hat{\rho}(\boldsymbol{\theta})} = \left( C[\hat{\rho}(\boldsymbol{\theta}), \hat{\mathbf{H}}, \hat{\mathbf{X}}] \right)_{kl}, \quad (24)$$

and we recover the moment matrix given in Eq. (1). We have assumed that $\Gamma[\hat{\rho}(\boldsymbol{\theta}), \hat{\mathbf{X}}]$ is invertible and $D[\hat{\rho}(\boldsymbol{\theta}), \hat{\mathbf{X}}]$ has rank $M$. This is usually the case for a suitable choice of the operators $\hat{\mathbf{X}}$ as is illustrated by our application to relevant examples of spin and continuous-variable systems. Rank deficiency of these matrices may indicate a redundancy in the information provided by the vector of measurement results that can be remedied by reducing the number of observables.

**Multiparameter shot-noise limit.** The classical precision limit of multiparameter distributed sensor networks, i.e., the multiparameter shot-noise limit, is defined as the maximal sensitivity that can be achieved by some optimally chosen classical probe state[17],

$$F_{\text{SN}}[\hat{\mathbf{H}}] := \max_{\hat{\rho}_{\text{cl}}} F_Q[\hat{\rho}_{\text{cl}}, \hat{\mathbf{H}}]. \quad (25)$$

The family of classical probe states $\hat{\rho}_{\text{cl}}$ depends on the system at hand. For a fixed number of particles, the system can effectively be described by discrete variables and a natural definition of classical states is given by particle-separable states[1,73]. Similarly, for continuous-variable systems we consider mixtures of coherent states as classical[9,74]. In the single-parameter theory, these families of classical states yield familiar expressions for the shot-noise limit, i.e., the $1/N$-scaling of the variance when $N$ is the number of particles, or the uncertainty of the vacuum state for homodyne measurements. These limits can be generalized to the multiparameter case, where the shot-noise matrix (Eq. (25)) is diagonal for locally encoded parameters[17]. The shot-noise limit for evolutions generated by $\hat{\mathbf{H}}$ is obtained from the quantum Cramér–Rao bound $\Sigma \geq (\mu F_Q[\hat{\rho}, \hat{\mathbf{H}}])^{-1}$ by considering the sensitivity of the optimal classical state:

$$\Sigma_{\text{SN}} = (\mu F_{\text{SN}}[\hat{\mathbf{H}}])^{-1}. \quad (26)$$

As a consequence of Eq. (3), we obtain that Eq. (11) holds for all classical states $\rho_{\text{cl}}$. The shot-noise limit in discrete-variable multimode interferometers is attained by the most sensitive particle-separable state $\hat{\rho}_{\text{p-sep}} = \sum_y p_y \hat{\rho}_1^{(y)} \otimes \cdots \otimes \hat{\rho}_N^{(y)}$, where $p_y$ is a probability distribution and the $\hat{\rho}_k^{(y)}$ are quantum states of particle $k$. Optimization over separable states leads to the shot-noise limit (Eq. (26)) defined in terms of a diagonal quantum Fisher matrix with diagonal elements given by the respective

numbers of particles in each mode[17]. Specifically, the classical sensitivity limit as a function of the accessible operators $\hat{\mathbf{J}}_\perp$ reads $F_{\text{SN}}[\hat{\mathbf{J}}_\perp] := \max_{\hat{\rho}_{\text{p-sep}}} F_Q[\hat{\rho}_{\text{p-sep}}, \hat{\mathbf{J}}_\perp] = \text{diag}(N_1, \dots, N_1, \dots N_M, \dots, N_M)$. For Hamiltonians $\hat{\mathbf{H}} = R \hat{\mathbf{J}}_\perp$ [recall Eq. (5)] that consist of linear combinations of the elements of $\hat{\mathbf{J}}_\perp$ this implies $F_{\text{SN}}[\hat{\mathbf{H}}] = R F_{\text{SN}}[\hat{\mathbf{J}}_\perp] R^T$. Further details are provided in Supplementary Note 3. Sensitivities beyond this limit can be achieved only by employing particle entanglement.

**Optimization of the phase-imprinting Hamiltonians.** To optimize the choice of $R$, i.e., the phase-imprinting Hamiltonians $\hat{\mathbf{H}}$, recall that the optimal moment matrix $\mathcal{M}_{\text{opt}}[\hat{\rho}, \hat{\mathbf{H}}, \hat{\mathbf{A}}]$ describes an $M \times M$ orthogonal projection of the larger $L \times L$ matrix $\tilde{\mathcal{M}}[\hat{\rho}, \hat{\mathbf{A}}]$. The eigenvectors and eigenvalues of $\mathcal{M}_{\text{opt}}[\hat{\rho}, \hat{\mathbf{H}}, \hat{\mathbf{A}}]$ both depend on the $M \times L$ matrix $R$. First, we notice that the basis of $\mathcal{M}_{\text{opt}}[\hat{\rho}, \hat{\mathbf{H}}, \hat{\mathbf{A}}]$ can be chosen at will by orthogonal transformations of the generating Hamiltonians: For any orthogonal $M \times M$ matrix $O$, we obtain

$$\mathcal{M}_{\text{opt}}[\hat{\rho}, O\hat{\mathbf{H}}, \hat{\mathbf{A}}] = OR\tilde{\mathcal{M}}[\hat{\rho}, \hat{\mathbf{A}}]R^T O^T. \quad (27)$$

Replacing $\hat{\mathbf{H}}$ by $O\hat{\mathbf{H}}$ does not affect the optimal measurement observables $\hat{\mathbf{X}}_{\text{opt}}$ since $O$ can be compensated by the matrix $G$ in Eq. (9) for $K = M$. Second, the eigenvalues of $\mathcal{M}_{\text{opt}}[\hat{\rho}, \hat{\mathbf{H}}, \hat{\mathbf{A}}]$ are determined by the $M$-dimensional support of $R$, which is spanned by $M$ out of the $L$ eigenvectors of $\tilde{\mathcal{M}}[\hat{\rho}, \hat{\mathbf{A}}]$. Since the basis of $\mathcal{M}_{\text{opt}}[\hat{\rho}, \hat{\mathbf{H}}, \hat{\mathbf{A}}]$ can be arbitrarily chosen via $O$, optimality of the $R$ is determined by the spectrum of $\mathcal{M}_{\text{opt}}[\hat{\rho}, \hat{\mathbf{H}}, \hat{\mathbf{A}}]$. We consider the parameter encoding optimal if $R$ projects onto the subspace corresponding to the $M$ largest eigenvalues of $\tilde{\mathcal{M}}[\hat{\rho}, \hat{\mathbf{A}}]$. For the common case of a shot-noise matrix $F_{\text{SN}}[\hat{\mathbf{H}}]$ that is proportional to the $M$-dimensional identity matrix, the same $R$ that is optimal for $\mathcal{M}_{\text{opt}}[\hat{\rho}, \hat{\mathbf{H}}, \hat{\mathbf{A}}]$ is also optimal for $\Xi^2_{\text{opt}}[\hat{\rho}, \hat{\mathbf{H}}, \hat{\mathbf{A}}]$. Similarly to the optimization over $\hat{\mathbf{X}}$, the phase-imprinting Hamiltonians $\hat{\mathbf{H}}$ must be constrained to physically implementable evolutions.

**Continuous-variable squeezing matrix.** Families of phase-space operators can be constructed from $\hat{\mathbf{q}}$ by means of a canonical transformation $O$ as $O\hat{\mathbf{q}}$. Canonical mode transformations are described by $2M \times 2M$ orthogonal symplectic matrices $O$ satisfying both $O^{-1} = O^T$ and $O\Omega O^T = \Omega$[11–13]. Notice that the elements of $O\hat{\mathbf{q}}$ are in general nonlocal linear combinations of those of $\hat{\mathbf{q}}$, but they follow the same commutation relations.

To discuss the problem of estimating $M$ parameters encoded by the local generators $\hat{\mathbf{H}} = R\hat{\mathbf{q}}$, we choose $R = P_M O$. Here the $M \times 2M$ projector $P_M$ onto canonical basis vectors with even labels picks a single operator (some linear combination of $\hat{x}$ and $\hat{p}$) from each of the local modes in $O\hat{\mathbf{q}}$, and $O$ is an orthogonal symplectic matrix. This condition ensures that all generators commute: using the condition $R = P_M O$, we find $\tilde{C}[\hat{\rho}, \hat{\mathbf{H}}] = \frac{1}{2} R\Omega R^T = \frac{1}{2} P_M O\Omega O^T P_M^T = \frac{1}{2} P_M \Omega P_M^T = 0_M$, and analogously, $\tilde{C}[\hat{\rho}, \hat{\mathbf{X}}] = 0_M$. Since $RR^T = 1_M$, this further implies that the shot-noise limit does not depend on the choice of generators, i.e., $F_{\text{SN}}[\hat{\mathbf{H}}] := \max_{\hat{\rho}_{\text{cl}}} F_Q[\hat{\rho}_{\text{cl}}, \hat{\mathbf{H}}] = 1_M$ for all $\hat{\mathbf{H}} = R\hat{\mathbf{q}}$.

**Maximal gain due to mode entanglement.** We first notice that, if the squeezing level is identical in all modes, the squeezing matrix becomes proportional to the identity matrix that leaves no room for further optimizations, i.e., all strategies perform equally well. In general, the mode-local state with the diagonal squeezing matrix [$U = 1_M$ in Eq. (19)] yields an estimation uncertainty of

$$\mu \mathbf{n}^T \Sigma \mathbf{n} = \mathbf{n}^T \Xi^2_{\text{opt}}[|\Psi_0\rangle, \hat{\mathbf{H}}, \hat{\mathbf{q}}]\mathbf{n} = \sum_{k=1}^M n_k^2 e^{-2r_k}. \quad (28)$$

To identify the corresponding sensitivity limit in the presence of mode entanglement, we change the eigenvectors of the squeezing matrix by applying a passive transformation $\hat{U}_V$ to the state $|\Psi_0\rangle$. We limit ourselves to passive transformations, since we consider the amount of initial squeezing a fixed resource[61]. We show in Supplementary Note 4 that passive transformations are sufficient to produce arbitrary basis transformations of the squeezing matrix. Let us denote $\mathbf{n}_1 = \mathbf{n}$ and complete it to a basis $\{\mathbf{n}_k\}_{k=1}^M$. Choosing a transformation $\hat{U}_V$ that achieves $\Xi^2_{\text{opt}}[\hat{U}_V|\Psi_0\rangle, \hat{\mathbf{H}}, \hat{\mathbf{q}}] = \sum_{k=1}^M e^{-2r_k} \mathbf{n}_k \mathbf{n}_k^T$, where $r_1 \geq \dots \geq r_M$, we obtain

$$\mu \mathbf{n}^T \Sigma \mathbf{n} = \mathbf{n}^T \Xi^2_{\text{opt}}[\hat{U}_V|\Psi_0\rangle, \hat{\mathbf{H}}, \hat{\mathbf{q}}]\mathbf{n} = e^{-2r_1}, \quad (29)$$

which clearly leads to a better precision than (Eq. (28)) as long as the squeezing level is not identical in all modes. While Eq. (28) makes use of all quadratures and yields the average squeezing, weighted by the normalized coefficients $n_k^2$, Eq. (29) maps the maximally squeezed quadrature onto the relevant linear combination of parameters. In other words, we have rotated the state $\hat{U}_V|\Psi_0\rangle$ such that the smallest eigenvector of $\Xi^2_{\text{opt}}[\hat{U}_V|\Psi_0\rangle, \hat{\mathbf{H}}, \hat{\mathbf{q}}]$ is given by $\mathbf{n}$. Notice that, in order to achieve this mapping for a nonlocal $\mathbf{n}$, the state $\hat{U}_V|\Psi_0\rangle$ becomes mode entangled.

In order to identify the limits of both strategies for a given $\mathbf{n}$, we consider the optimal distribution of a finite total amount of squeezing that minimizes Eq. (28) or Eq. (29) for a fixed total average number of particles

$N = \sum_{k=1}^{M} \langle \hat{a}_k^\dagger \hat{a}_k \rangle_{|\Psi_0\rangle} = \sum_{k=1}^{M} \sinh^2 r_k$. The constrained minimization of Eq. (28) is done with the method of Lagrange multipliers: we write the Lagrange function $\mathfrak{L}(\mathbf{x},\lambda) = \sum_{k=1}^{M} \frac{n_k^2}{x_k} - \lambda \left[ \sum_{k=1}^{m} \left( \frac{x_k}{4} + \frac{1}{4x_k} \right) - \frac{M}{2} - N \right]$, where $x_k = e^{2r_k}$. The solution of the set of $M+1$ equations $\frac{d\mathfrak{L}(\mathbf{x},\lambda)}{d\lambda} = 0$ for $k = 1, \ldots, M$ and $\frac{d\mathfrak{L}(\mathbf{x},\lambda)}{dx_k} = 0$ gives $n_k^2 = (\lambda/4)(x_k^2 - 1)$. Summing over $k$ and imposing $\sum_{k=1}^{M} n_k^2 = 1$, we find

$$\frac{e^{4r_k} - 1}{\sum_{k=1}^{M}(e^{4r_k} - 1)} = n_k^2, \qquad (30)$$

whose solution gives the optimal squeezing parameters $r_k$.

Clearly, the mode-entangled sensitivity (Eq. (29)) is optimized by concentrating all available squeezing into the initial mode that will be mapped by $\hat{U}_V$ onto the optimal nonlocal mode, characterized by $\mathbf{n}$, leading to

$$(\Delta\theta_{\text{m-ent}})^2 = e^{-2r'}, \qquad (31)$$

where $\sinh^2 r' = \sum_{k=1}^{M} \sinh^2 r_k$ for the conservation of the total average particle number.

In the following, let us consider, for simplicity, the estimation of an equally weighted linear combination of all parameters, i.e., $n_k^2 = 1/M$ for $k = 1, \ldots, M$. This implies that all the $r_k \equiv r$ are identically chosen and $(\Delta\theta_{\text{m-sep}})^2 = e^{-2r}$, where $r = \text{arcsinh}\sqrt{N/M}$. The entanglement-enabled noise suppression factor is given by $(\Delta\theta_{\text{m-ent}})^2/(\Delta\theta_{\text{m-sep}})^2 = e^{-2r'}/e^{-2r}$. In the case $r = 0$ (that also implies $r' = 0$), we have $(\Delta\theta_{\text{m-ent}})^2/(\Delta\theta_{\text{m-sep}})^2 = 1$: the mode-entangled and mode-separable strategies perform equally well. When $r \ll 1/\sqrt{M}$, we can approximate $r' \approx \sqrt{M}r$ (recall that $\sinh^2 r \approx r^2 + O(r^3)$) and obtain $(\Delta\theta_{\text{m-ent}})^2/(\Delta\theta_{\text{m-sep}})^2 \approx e^{-2(\sqrt{M}-1)r}$. When $r \gg 1$ (that also implies $r' \gg 1$), we have $e^{-2r'} \approx M e^{-2r}$.

## Data availability

All relevant data are available from the authors.

## Code availability

Source codes of the plots are available from the corresponding author upon request.

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

## Acknowledgements

This work was supported by the LabEx ENS-ICFP: ANR-10-LABX-0010/ANR-10-IDEX-0001-02 PSL* and the European Commission through the QuantERA ERA-NET Cofund in Quantum Technologies project "CEBBEC". The authors acknowledge financial support from the European Union's Horizon 2020 research and innovation program – Qombs Project, FET Flagship on Quantum Technologies grant no. 820419.

## Author contributions

M.G., A.S., and L.P. contributed to all aspects of this work.

## Competing interests

The authors declare no competing interests.
