## [Peer Review File · Nature Communications]

Reviewers' Comments:

Reviewer #1:

Remarks to the Author:

Please see the attached PDF file

Reviewer #2:

Remarks to the Author:

The manuscript by Gessner et al. is a contribution to multiple-parameter quantum metrology. It proposes to use first and second moments of a compatible set of measurement operators (one for each parameter) in order to obtain bounds on the mean squared error that are achievable in the large-sample limit. Since achievability of quantum Cramér-Rao bounds is not always possible in multiparameter estimation, the focus on achievable bounds is welcome. The authors apply the bounds they develop to the estimation of multiple phase shifts in spin systems, and to multimode displacement parameters in optical systems, which should be of interest to experimenters in the field.

The conclusions of the paper are interesting and correct as far as I can tell. However, I would like to note that their bound connecting the classical Fisher information matrix to the moment matrix (the first bound in Eq. (3)) is known in the classical statistics literature: see, e.g. the paper by Stein, Mezghani, Nossek "A Lower Bound for the Fisher Information Measure," IEEE Signal Processing Letters (Volume: 21 , Issue: 7 , July 2014).

However, their emphasis on achievability of the bound in the large-sample limit (by applying the central limit theorem) and the optimization of the measurement operators to maximize the lower bound are novel as far as I can tell. Citing the above result should help shorten some of the authors' proofs.

With the above caveats, I think the paper is suitable for publication.

Reviewer #3:

Remarks to the Author:

The use of quantum states to enhance the performance of a range of metrological schemes is rapidly emerging as an exciting new quantum technology. Much of the early focus has been on how a single parameter can be measured with enhanced precision, using tools such as the quantum Fisher information and associated Cramer-Rao bound. Intriguingly, the best approach, at least from a practical point of view, has been to use squeezed states and is close to the one originally envisioned by Caves in the 1980s. This is now used in experiments such as LIGO. As the field has matured, attention has turned to how multiple parameters can be simultaneously measured. This is of great interest not just because many real-world problems involve more than one parameter but also the multiparameter problem can be fundamentally different from the single parameter one. It is known, for example, that measuring many parameters simultaneously in a quantum context can give an advantage over measuring each one individually. Things also get much more complicated when the generators associated with each of the parameters do not commute. In this case there are interesting trade-offs in the precision achievable.

Theoretical work in this area has looked at networks of sensors and functions of parameters and has made use of tools such as the Holevo Cramer-Rao bound which is asymptotically attainable in the multiparameter case. The problem is that, while theoretically important, this bound can be difficult to calculate let alone achieve in practice. So there is a case for finding other bounds that might not be quite as good but have practical advantages (e.g. Tsang arXiv:1911.08359). The present manuscript takes a similar approach and, given that squeezing has been so successful for

single parameters, asks whether a multiparameter version of squeezing might be similarly successful in that regime.

The manuscript develops the theory clearly and accurately and presents results that will be of interest to the community. I have no doubt that this paper should be published, however I am not yet sure whether it fulfils the special requirements for publication in Nature Communications. My main reason for this is that the result is not as general as is claimed. The abstract says: "...we derive the general operational concept of multiparameter squeezing, identifying metrologically useful states and optimal estimation strategies for arbitrary states and systems." In fact some important assumptions have been made that means this is not optimal for arbitrary states and systems. In particular it has been assumed that only Gaussian states are used – this is well-motivated physically but does mean that the result is not general. Secondly, and most importantly, it is assumed that the generators associated with all the parameters commute. This heavily restricts the class of problems that it addresses and arguably excludes the most interesting quantum metrology scenarios.

As I say, I think this initial paper on multiparameter squeezing is very nice and I would not expect it to have to answer all these questions. However I raise them as they are closely associated with the importance and reason for doing this work. In the final paragraph it is said that by extending the set of accessible observables, the technique may be extended to arbitrary non-Gaussian multimode states. I would like to see some more justification for how/why this might work and maybe also some comment on the non-commuting case. This would show a clear step forward over the current scenario where we have general (but hard and impractical) theory versus more accessible (but restricted) approaches. If it can be argued that the methods in the present manuscript could be extended to other more general cases while retaining practical advantages then I think there would be a good case for publication in Nature Communications.

Reviewer #1 (Remarks to the Author):

Multiparameter estimation is now one of the central topics in the study of theoretical quantum metrology, and due to its high computational complexity and difficulty in the construction of optimal measurements, the developments on novel theoretical tools should always be welcomed. In this paper, a Gaussian multivariate distribution is considered. By calculating its (classical) Fisher information and keeping the corresponding leading term, the covariance matrix is then expressed by the moment matrix. The authors then show the relations of the moment matrix with the classical and quantum Fisher information, and one specific optimal observable that can saturate the classical Fisher information. With the given moment matrix and the multiparameter shot-noise limit, the authors define a so-called squeezing matrix to quantify the quantum gain in a moment-based multiparameter estimation protocol. Several scenarios have been discussed as the applications. The paper is indeed interesting, well written and the given squeezing matrix may do have a potential to have a wide applications in quantum metrology and quantum information. Hence, I am happy to recommend this paper to be published in Nature Communications once the following concerns are considered.

We thank the Reviewer for the positive assessment of our manuscript and for recommending it for publication in Nature Communications. In the following we address the Referee's comments in detail:

1. It seems like that the performance of nonlocal spin squeezing can only outperform the local ones for a very small χt . I think the authors should discuss the feasibility of the entire scheme to be finished within this short time scale (compared to $1/\chi$) in experiments, and either show that this gain can indeed be realized in practice or point out the realistic difficulties.

The parameter χt describes the time that is required in the preparation of the optimal state using interaction between the atoms, which is described by the nonlinear Hamiltonian χJ_z^2 . Short time scales (namely small χt) are indeed easier (and most relevant) for experiments in order to avoid decoherence. In the case of single-parameter estimation, sensitivities below the shot noise limit have been achieved for small χt in different systems (see the review [1]). Notice that further experimental times for the actual phase measurement protocol are typically implemented by pulses much shorter than χt . Furthermore, non-local spin squeezing can be experimentally created with techniques similar to that of Refs. [47-49] where a cloud of interacting atoms in a Bose-Einstein condensate is split after a finite interaction time that can be described by the nonlinear evolution discussed in our manuscript. Therefore, the experimental demonstration that nonlocal spin squeezing can outperform both the shot-noise limit and local spin squeezing strategies is feasible within the current technology.

We have added a sentence below Eq. (14) to clarify the role of χt .

2. In the discussion of maximum enhancement due to mode entanglement, I suggest the authors not only consider a specific combination, but also a real multiparameter estimation (at least with two parameters) and compare the performances of mode separable and mode-entangled strategies.

We would like to point out that we always compare the performance of real multiparameter estimation measurements by determining their full sensitivity matrix. However, comparing two alternative multiparameter strategies is complicated by the fact that it may not always be possible to quantify the differences between their sensitivity matrices in terms of a single number. For this reason, it is customary to identify suitable scalar figures of merit: While many studies focus on the

trace of the sensitivity matrix, we focus on arbitrary linear combinations of parameters because it also contains information about the correlations that are ignored by the trace. The information provided by all possible linear combinations is indeed equivalent to the full multiparameter estimation sensitivity.

Stimulated by the Reviewer's comment, we have added a sentence clarifying this point at the beginning of the section on the maximum enhancement due to mode entanglement.

3. The Hamiltonians in \mathbf{H} (and observables in \mathbf{X}) considered here require the commutation with each other, which is of course a reasonable scenario. However, there also exist plenty of cases that the parameters are encoded with noncommutative generators. Hence, I suggest the authors give some comments on possibility or the difficulties of this method to be generalized into that case. Moreover, in the discussion around Eq. (6), personally I think the $\mathfrak{su}(n)$ generators would be a possible and interesting example. I suggest the authors consider to take it in.

We thank the Referee for this comment that has stimulated us to clarify the general conditions under which our results hold. We have extended the discussion and shown that the initial assumption of commuting Hamiltonians \mathbf{H} is not necessary: the theory extends to non-commutative operators as well. We also clarify that the commutativity of operators \mathbf{X} is not a necessary but a practical choice: the measurement of non-commuting observables unavoidably requires the repetition of the experiments with identical copies of the probe state, which represents a loss of resources.

Furthermore, to illustrate the wider scope of our theory, we have included in the main text an example for the non-commuting scenario. Following the Referee's advice, we consider the application of our method to the estimation of two angle of a $SU(2)$ with non-commuting generators and using a Twin-Fock state as probe, which provides Heisenberg scaling $\sim N^2$ of sensitivity for the joint estimation of both angles. This is indeed a very interesting example where, despite the non-commutativity of the generators, we are able to identify optimal (commuting) measurement observables for which our moment matrix saturates the quantum Fisher information matrix and is thus an optimal phase estimation method.

Generally speaking, our formalism can be applied to non-commuting scenarios and provides saturable sensitivity bounds for any choice of measurement observables. However, we cannot make general statements about the quality of these bounds in general, since in non-commuting scenarios, the achievable ultimate quantum limits of sensitivity are unknown, and thus, we have no reliable benchmark for comparison. The additional example shows that the sensitivity achieved by our method can be as large as the ultimate multiparameter quantum limit.

4. A typo: I think the first equation in the Supplementary Material should be Eq. (1.1), not the second one, according to the contextuality.

We thank the Referee for pointing this out. As suggested, we labelled the first equation in the Supplementary Material by Eq. (1.1).

Reviewer #2 (Remarks to the Author):

The manuscript by Gessner et al. is a contribution to multiple-parameter quantum metrology. It proposes to use first and second moments of a compatible set of measurement operators (one for each parameter) in order to obtain bounds on the mean squared error that are achievable in the large-sample limit. Since achievability of quantum Cramér-Rao bounds is not always possible in

multiparameter estimation, the focus on achievable bounds is welcome. The authors apply the bounds they develop to the estimation of multiple phase shifts in spin systems, and to multimode displacement parameters in optical systems, which should be of interest to experimenters in the field.

The conclusions of the paper are interesting and correct as far as I can tell. However, I would like to note that their bound connecting the classical Fisher information matrix to the moment matrix (the first bound in Eq. (3)) is known in the classical statistics literature: see, e.g. the paper by Stein, Mezghani, Nossek "A Lower Bound for the Fisher Information Measure," IEEE Signal Processing Letters (Volume: 21 , Issue: 7 , July 2014).

However, their emphasis on achievability of the bound in the large-sample limit (by applying the central limit theorem) and the optimization of the measurement operators to maximize the lower bound are novel as far as I can tell. Citing the above result should help shorten some of the authors' proofs.

With the above caveats, I think the paper is suitable for publication.

We thank the Reviewer for the positive comments on our manuscript and for recommending it for publication. We further thank the Reviewer for pointing our attention to the above-mentioned reference from classical statistics. Its content is in fact a derivation of Eq. (2.17) in our Supplementary Material. However, since the proof follows in a single line, we decided not to remove it to keep our manuscript as self-contained and readable as possible. We have added citations to the suggested reference in the main text and the Supplementary Material.

As pointed out by the Reviewer, our results go much further than the results of Stein et al. in several respects:

- (i) we discuss this (classical) inequality in the case of a quantum measurement and identify the observables that achieve saturation,
- (ii) we derive a quantum version of this bound and discuss the conditions for saturation,
- (iii) we develop an analytical optimization method to identify the accessible observables that maximize this lower bound, and
- (iv) we provide an operational method that achieves this multiparameter sensitivity and use it to detect quantum correlations from the squeezing of the measured observables.

Reviewer #3 (Remarks to the Author):

The use of quantum states to enhance the performance of a range of metrological schemes is rapidly emerging as an exciting new quantum technology. Much of the early focus has been on how a single parameter can be measured with enhanced precision, using tools such as the quantum Fisher information and associated Cramer-Rao bound. Intriguingly, the best approach, at least from a practical point of view, has been to use squeezed states and is close to the one originally envisioned by Caves in the 1980s. This is now used in experiments such as LIGO. As the field has matured, attention has turned to how multiple parameters can be simultaneously measured. This is of great interest not just because many real-world problems involve more than one parameter but also the multiparameter problem can be fundamentally different from the single parameter one. It is known, for example, that measuring many parameters simultaneously in a quantum context can give an advantage over measuring each one individually. Things also get much more complicated when the generators associated with each of the parameters do not commute. In this case there are interesting trade-offs in the precision achievable.

Theoretical work in this area has looked at networks of sensors and functions of parameters and has made use of tools such as the Holevo Cramer-Rao bound which is asymptotically attainable in the multiparameter case. The problem is that, while theoretically important, this bound can be difficult to calculate let alone achieve in practice. So there is a case for finding other bounds that might not be quite as good but have practical advantages (e.g. Tsang arXiv:1911.08359). The present manuscript takes a similar approach and, given that squeezing has been so successful for single parameters, asks whether a multiparameter version of squeezing might be similarly successful in that regime.

The manuscript develops the theory clearly and accurately and presents results that will be of interest to the community. I have no doubt that this paper should be published, however I am not yet sure whether it fulfils the special requirements for publication in Nature Communications. My main reason for this is that the result is not as general as is claimed. The abstract says: "...we derive the general operational concept of multiparameter squeezing, identifying metrologically useful states and optimal estimation strategies for arbitrary states and systems." In fact some important assumptions have been made that means this is not optimal for arbitrary states and systems. In particular it has been assumed that only Gaussian states are used – this is well-motivated physically but does mean that the result is not general. Secondly, and most importantly, it is assumed that the generators associated with all the parameters commute. This heavily restricts the class of problems that it addresses and arguably excludes the most interesting quantum metrology scenarios.

As I say, I think this initial paper on multiparameter squeezing is very nice and I would not expect it to have to answer all these questions. However I raise them as they are closely associated with the importance and reason for doing this work. In the final paragraph it is said that by extending the set of accessible observables, the technique may be extended to arbitrary non-Gaussian multimode states. I would like to see some more justification for how/why this might work and maybe also some comment on the non-commuting case. This would show a clear step forward over the current scenario where we have general (but hard and impractical) theory versus more accessible (but restricted) approaches. If it can be argued that the methods in the present manuscript could be extended to other more general cases while retaining practical advantages then I think there would be a good case for publication in Nature Communications.

We thank the Reviewer for the positive assessment of our manuscript. We have focused our manuscript on Gaussian states and commuting generators mainly because this is currently the most relevant experimental scenario. Nevertheless, inspired by the Referee's suggestion, we have clarified the general conditions under which our results hold: Our multiparameter method of moments actually does not require the commutativity of the generating Hamiltonians. Clarifying this point has certainly increased the quality and impact of our work. To highlight the wider scope of our method, we also included a new example, before the "Discussion and Conclusion" section, discussing the application of our estimation method to a non-Gaussian state (the twin-Fock state) and non-commuting generators (of $SU(2)$). We show that our method saturates the quantum Fisher matrix and is thus an optimal estimation method.

Let us discuss the two suggested generalizations by the Referee separately:

i) Non-Gaussian states. The possible generalization of our approach to non-Gaussian states is pointed out by the fact that, for an optimally chosen observable, we saturate the classical Fisher matrix (for any basis) [see Eq.(4) in our manuscript and additional details in Supplemental Note II D]. The Fisher matrix is known to capture all relevant features of highly sensitive non-Gaussian states and to fully describe their multiparameter sensitivity. For Gaussian states this saturation is achieved by first and second moments of simple, linear measurement observables. This leads to some of the

most important practical approaches that are at the heart of the examples discussed in our manuscript. For non-Gaussian states, our approach generalizes straightforwardly by using measurements that access information contained in higher-order moments (beyond averages and variances) that can be made accessible by the measurement of nonlinear observables. Certainly, accessing higher moments is more challenging experimentally, but it is the unavoidable price to pay when using non-Gaussian states.

ii) Non-commuting generators. The initial assumption of commuting Hamiltonians H is not necessary: the theory extends to non-commutative operators as well (the commutativity of Hamiltonians never entered explicitly in the derivation of our equations). There are two main limitations posed by non-commuting Hamiltonians:

-) One is the lack of an explicit expression for the shot noise limit. While the shot noise can be formally defined as the maximum sensitivity achievable with classical states, an explicit expression cannot be given, in general, in terms of the quantum Fisher matrix. This is because the quantum Cramer-Rao bound cannot be saturated in general (while it can always be saturated for the commuting scenario). This is an unsolved problem in the literature and is beyond the scope of this manuscript. Even though the moment-based sensitivity is always well defined, this limitation does not allow us to give an explicit expression for the squeezing matrix in the non-commuting case. Instead we provide a formal definition which can be used once the shot noise is defined.

-) The other limitation is the fact that optimal measurement observables are not known, in general, in the non-commuting scenario with mixed states as probe. This long-standing problem remains open since the pioneering approaches by Helstrom and Holevo in 60s, and, in all generality, it is beyond the scope of this manuscript. We provide a method that by construction leads to saturable multiparameter sensitivity bounds for an arbitrary set of observables. We further provide methods for the optimization of measurement observable as linear combination of a given set of (experimentally) accessible operators. While our methods are experimentally relevant and effective, we are at this point unable to tell whether our strategy qualifies as the absolute optimum in general multiparameter scenarios, since we have no ultimate quantum bound in general to compare to.

To summarize, (i) we can show generally that our method (including optimization results) can be extended to non-Gaussian observables and states. (ii) We can treat non-commuting generators using our method and the additional example shows that at least in some cases a set of commuting observables can lead to a multiparameter sensitivity that saturates the ultimate quantum limit. The systematic identification of commuting, optimal measurement observables for general cases are presently unknown and lie beyond the scope of this manuscript. The new example clearly shows that the extension of our results to other more general cases is well motivated and most practical advantages can be retained in these scenarios.

Further changes:

-) We agree that the Holevo bound pointed out by the Referee is an interesting alternative to the Cramer-Rao approach discussed in our manuscript, especially in non-commuting scenarios. We have mentioned this aspect with a corresponding remark in the introduction. We have added the reference mentioned by the Referee (in particular we cite an updated version that was merged with work by other authors).

-) The phrase "arbitrary states and systems" in the abstract holds in the context of distributed sensor networks, which generally implies commuting generators. There, we can treat arbitrary, even non-Gaussian states, in arbitrary-dimensional systems. Certainly "arbitrary states and systems" is misleading if one considers non-commuting generators. Therefore, in order to avoid the risk of misinterpretations, which was pointed out by the Reviewer, we have removed this phrase.

We hope that the additional example and further clarifications have adequately addressed the Reviewer's question.

Full list of Changes

Main text:

- We removed the phrase on “arbitrary states and systems” in the abstract.
- We added a remark in the introduction on the Holevo bound with additional references [38-42].
- The multiparameter method of moments and general principle of the squeezing matrix is presented more generally without assuming commuting Hamiltonians; potential of non-Gaussian measurements is explained more clearly.
- In response to Reviewer 2's remark, we added a citation to Ref. [56] after Eq. (3).
- In response to Reviewer 1's question, we added a comment on the role of the interaction time.
- We added a remark on multiparameter sensitivity in the discussion of the maximal enhancement due to mode entanglement, responding to Reviewer 1's question.
- We added a section where the new example on SU(2) estimation with a Twin-Fock state is discussed.
- We added a remark in the conclusions that applications with non-commuting scenarios are in principle possible but have not been explored in full detail.
- In agreement with the updated presentation in the main text we made minor modifications in the methods section to ensure consistency.
- In general, we made small modifications in the text to save space for the new example and to correct minor typos.

Supplementary Material:

- Minor modifications to ensure consistency with the updated presentation of the main text.
- We added a section demonstrating explicitly the possibility to saturate the classical Cramer Rao bound with our method through projective measurements, which we discuss in the broader context of reaching the ultimate quantum limit with non-commuting generators.

Reviewers' Comments:

Reviewer #1:

Remarks to the Author:

The current form of the manuscript has included all my concerns. Hence, now I can recommend it to be published in Nature Communications.

Reviewer #2:

Remarks to the Author:

I think the current revision addresses my earlier concerns, and is suitable for publication.

Reviewer #3:

Remarks to the Author:

I think that the authors have done an excellent job of responding to my comments. They have now satisfied me about their claims of generality and I think that the new example was particularly helpful on that front. On that basis, I am happy to recommend publication.